# Injection with *Toxoplasma gondii* protein affects neuron health and survival

Oscar A Mendez[1], Emiliano Flores Machado[2], Jing Lu[3], Anita A Koshy[2,4,5]*

[1]Graduate Interdisciplinary Program in Neuroscience, University of Arizona, Tucson, United States; [2]BIO5 Institute, University of Arizona, Tucson, United States; [3]College of Nursing, University of Arizona, Tucson, United States; [4]Department of Immunobiology, University of Arizona, Tucson, United States; [5]Department of Neurology, University of Arizona, Tucson, United States

**Abstract** *Toxoplasma gondii* is an intracellular parasite that causes a long-term latent infection of neurons. Using a custom MATLAB-based mapping program in combination with a mouse model that allows us to permanently mark neurons injected with parasite proteins, we found that *Toxoplasma*-injected neurons (TINs) are heterogeneously distributed in the brain, primarily localizing to the cortex followed by the striatum. In addition, we determined that cortical TINs are commonly (>50%) excitatory neurons (FoxP2[+]) and that striatal TINs are often (>65%) medium spiny neurons (MSNs) (FoxP2[+]). By performing single neuron patch clamping on striatal TINs and neighboring uninfected MSNs, we discovered that TINs have highly aberrant electrophysiology. As approximately 90% of TINs will die by 8 weeks post-infection, this abnormal physiology suggests that injection with *Toxoplasma* protein—either directly or indirectly—affects neuronal health and survival. Collectively, these data offer the first insights into which neurons interact with *Toxoplasma* and how these interactions alter neuron physiology in vivo.

## Introduction

A select number of highly divergent microbes (e.g., measles virus, *Toxoplasma gondii*) naturally cause infections of neurons within the central nervous system (CNS). Most of these viruses cause debilitating disease or death, whereas alphaherpes viruses and the eukaryotic parasite *Toxoplasma* primarily establish persistent, relatively quiescent neuronal infections in the immunocompetent host (*Dubey, 2009*). While neuron-alphaherpes virus interactions have been studied for decades—leading to a mechanistic understanding of herpes virus latency in neurons and the development of novel tools for circuit tracing—our understanding of neuron-*Toxoplasma* interactions is just beginning (*Ugolini, 1995*; *Wickersham et al., 2007*).

*Toxoplasma* is a ubiquitous intracellular parasite that can infect a wide range of warm-blooded hosts, including birds, rodents, and humans. In most immunocompetent hosts, *Toxoplasma* establishes a long-term, asymptomatic infection in specific organs. In humans and rodents, the CNS is a major organ of persistence and is the organ most affected by symptomatic disease in immunocompromised patients (*Remington and Cavanaugh, 1965*; *Luft and Remington, 1992*; *Dubey, 2009*; *Neu et al., 2015*). Using a Cre-based mouse model in which host cells injected with *Toxoplasma* protein permanently express a green fluorescent protein (GFP), our group showed that *Toxoplasma* predominantly interacts with neurons, not glia, in vivo (*Koshy et al., 2012*; *Cabral et al., 2016*). In addition, we determined that neurons injected with *Toxoplasma* protein outnumber cysts by over 20-fold (*Koshy et al., 2012*), suggesting that far more neurons interact with *Toxoplasma* than are persistently infected.

Here, we continue to harness this Cre-based model to extend our understanding of the neuron-*Toxoplasma* interface. We leverage the high numbers of *Toxoplasma*-injected neurons (TINs) and

*For correspondence:
akoshy@arizona.edu

Competing interests: The authors declare that no competing interests exist.

**eLife digest** *Toxoplasma gondii* is an intracellular parasite that infects the brain. Whereas most microbial infections of the brain result in severe illness or death, *Toxoplasma gondii* infections are usually asymptomatic. This is because the parasite has evolved the ability to exist within the brain by dampening the immune response. The parasite can therefore asymptomatically co-exist with its host for years – or even an entire lifetime. The strategy has proved so successful that up to one third of the world's population is now thought to be infected with *Toxoplasma gondii.*

While this persistence tends not to be a problem for most healthy individuals, dormant *Toxoplasma gondii* parasites can reactivate in individuals whose immune systems fail. This can result in life-threatening neurological disease. In pregnant women, *Toxoplasma gondii* parasites can also cross the placenta, which can trigger miscarriage or cause harmful disease in the newborn.

To develop treatments for these cases of symptomatic disease, we need to understand how the parasite hides from the immune system in asymptomatic individuals. Mendez et al. have therefore leveraged a mouse model in which neurons injected with *Toxoplasma gondii* proteins (*Toxoplasma*-injected neurons, or 'TINs') produce a green fluorescent protein. This enables the infected cells to be viewed under a microscope.

Examining the mouse brains revealed that most TINs were located in two specific regions: the cortex and the striatum. The cortex is the brain's outer layer of tissue. The striatum is a structure deep within the brain that helps regulate movement and responses to rewards. Both the cortex and the striatum contain different types of neurons. The results revealed that the proteins from the parasite were spread roughly equally among the various cell types, rather than targeting a specific subtype of neuron.

Neurons close to TINs had slightly abnormal electrical activity, whereas the TINs themselves had highly abnormal activity. By eight weeks post-infection, however, the number of TINS had fallen by around 90%. This suggests that many neurons containing *Toxoplasma* protein are sick and dying, and that their altered electrical activity reflects this unhealthy state.

Understanding how *Toxoplasma* parasites persist in the brain has the potential to reveal new targets for treating symptomatic infections. It could even provide new possibilities for targeting the inflammation that drives many other neurological diseases. Harnessing this potential will require finding out why *Toxoplasma gondii* infects specific brain regions and why most neurons that directly interact with the parasite die.

the expression of GFP, which labels the full neuron, to carry out a systematic neuroanatomic mapping of TINs. These studies revealed that TINs are most commonly found within the cortex followed by the striatum and are rarely found in the cerebellum. Within the cortex and the striatum, we used immunofluorescent assays to determine if *Toxoplasma* preferentially interacts with specific neuron subtypes, finding instead that the lineage of TINs essentially mirrors regional neuron subtype abundance. Finally, we used single neuron patch clamping in ex vivo slices to compare the electrophysiology of striatal TINs to neighboring 'bystander' neurons (neurons within the same slice that were not injected with *Toxoplasma* proteins). These first-of-their-kind studies for any neurotropic microbe showed that TINs—but not bystander neurons—had dramatically altered electrophysiology. At 8 weeks post-infection (wpi), the number of TINs drops by approximately 90%, suggesting this abnormal physiology reflects an unhealthy cellular state. Collectively, these data offer the first insights into which neurons interact with *Toxoplasma* and how these interactions—either directly or indirectly—alter neuron physiology and survival in vivo.

## Results

### TINs are enriched in the cortex, irrespective of infecting *Toxoplasma* strain

Prior work in AIDS patients and rodents suggested that *Toxoplasma* does not evenly distribute across the brain (*Post et al., 1983*; *Lang et al., 1989*; *Luft and Remington, 1992*; *Arendt et al., 1999*; *Porter and Sande, 1992*; *Strittmatter et al., 1992*; *Berenreiterová et al., 2011*;

*Evans et al., 2014*; *Neu et al., 2015*; *Dubey et al., 2016*). These studies have conflicting findings as to where *Toxoplasma* lesions or cysts (in rodent studies) are predominantly found, differences that might be driven by the small numbers of patients or rodents analyzed (*Berenreiterová et al., 2011*; *Evans et al., 2014*; *Dubey et al., 2016*; *Boillat et al., 2020*). Given these conflicting studies, we sought to neuroanatomically map the location of TINs, a population that includes both uninfected, injected neurons and actively infected neurons (*Koshy et al., 2012*). To accomplish this goal, we used our previously published semi-automated method (*Mendez et al., 2018*) which utilizes the Allen Institute Mouse Brain Atlas as a reference to quantify and map TINs quickly. This automation made it feasible to sample multiple brain sections per mouse across multiple cohorts of mice infected with either of two genetically distinct *Toxoplasma* strains (type II— Prugniaud, type III— CEP) that express a *Toxoplasma*:Cre fusion protein and mCherry (*Koshy et al., 2010*; *Koshy et al., 2012*; *Christian et al., 2014*). For simplicity, from here forward, we will refer to these strains as II-Cre and III-Cre. We quantified TINs in 16–19 brain sections/mouse across 32 regions identified in the Allen Institute Mouse Brain Atlas. Data points were grouped into 12 major regions. For both II-Cre and III-Cre infected mice, the cortex contained the highest average number of TINs, followed by the striatum (*Figure 1A,B*; *Figure 1—figure supplement 1*). In general, III-Cre infected mice showed a higher overall number of TINs compared to II-Cre infected mice (average of 247.5 III-Cre versus 216.7 II-Cre). The increase in absolute numbers in III-Cre infected mice was most pronounced in the cortex (*Figure 1A,B*). Most other brain regions, including the cerebellum, contained relatively few TINs (*Figure 1A,B*).

As the cortex and striatum encompass a large proportion of the brain, the high number of TINs in these areas might be driven by relative size. To test this possibility, we produced an enrichment index using the following equation: $\frac{\left(\frac{Rtin}{Ttin}\right)}{Rsize}$, where *Rtin* is the TIN count for a given region (e.g., cortex), *Ttin* is the total TIN count from an individual mouse, and *Rsize* is the total area percentage of a specific region (e.g., % of brain that is cortex) (see *Supplementary file 1* for area percentage for each brain region). Our method of sectioning does not allow us to collect the smaller sections of the cerebellum, therefore we only included the cerebellar areas from sections 10-16 of the Allen atlas for normalization (consistent with the regions of the cerebellum which were analyzed). To align our sections with the Allen Institute sections, we also excluded the olfactory bulb (*Mendez et al., 2018*). Even with these exclusions, 85% of the original Allen atlas area was used. If the number of TINs in a region is proportional to the size of that region, the normalization index will approximate 1, while regions enriched for TINs will have indices > 1, and regions relatively devoid of TINs will have indices < 1. Using this normalization, we determined that the cortex was the only region which was significantly enriched for TINs in both II-Cre and III-Cre infected mice (index score 1.4 ± 0.1 and 2.2 ± 0.2, respectively) (*Figure 1C, D*). For both groups of infected mice, within the cortex, the somatosensory, motor, and visual cortices had the highest enrichment score (*Figure 1—figure supplement 2*). Finally, two regions—the cerebellum and the white matter tracts—showed a significant lack of enrichment in both II-Cre and III-Cre infected mice (*Figure 1C, D*). The lack of TINs in white matter tracts is expected given that these tracts are primarily made up of neuron axons and thus have a paucity of neuron nuclei, the cellular component by which neurons are anatomically mapped. *Supplementary file 2* has the full list of enrichment scores and statistical analysis for the 12 listed regions.

In summary, these data show that, irrespective of infecting *Toxoplasma* strain, TINs are commonly found in the cortex followed by the striatum and relatively rarely found in the cerebellum. While TINs localization in the striatum is consistent with its size, the enrichment of TINs in the cortex and lack of TINs in the cerebellum are not accounted for by the relative size of these regions.

## TINs rarely co-localize with markers of inhibitory neurons

As the cortex and the striatum had the highest number of TINs, we decided to use these areas to determine if *Toxoplasma* targeted a specific neuron population. Neurons can be classified in many ways, including by neurotransmitter expression, physiology, and morphology. The simplest way is to classify neurons is as inhibitory or excitatory. Given that prior work suggested that *Toxoplasma* infection might specifically affect inhibitory neurons (*Brooks et al., 2015*), we first examined how often TINs co-localized with two common markers for inhibitory interneurons, calbindin and parvalbumin. We chose these makers rather than the pan-inhibitory marker glutamate decarboxylase (GAD)

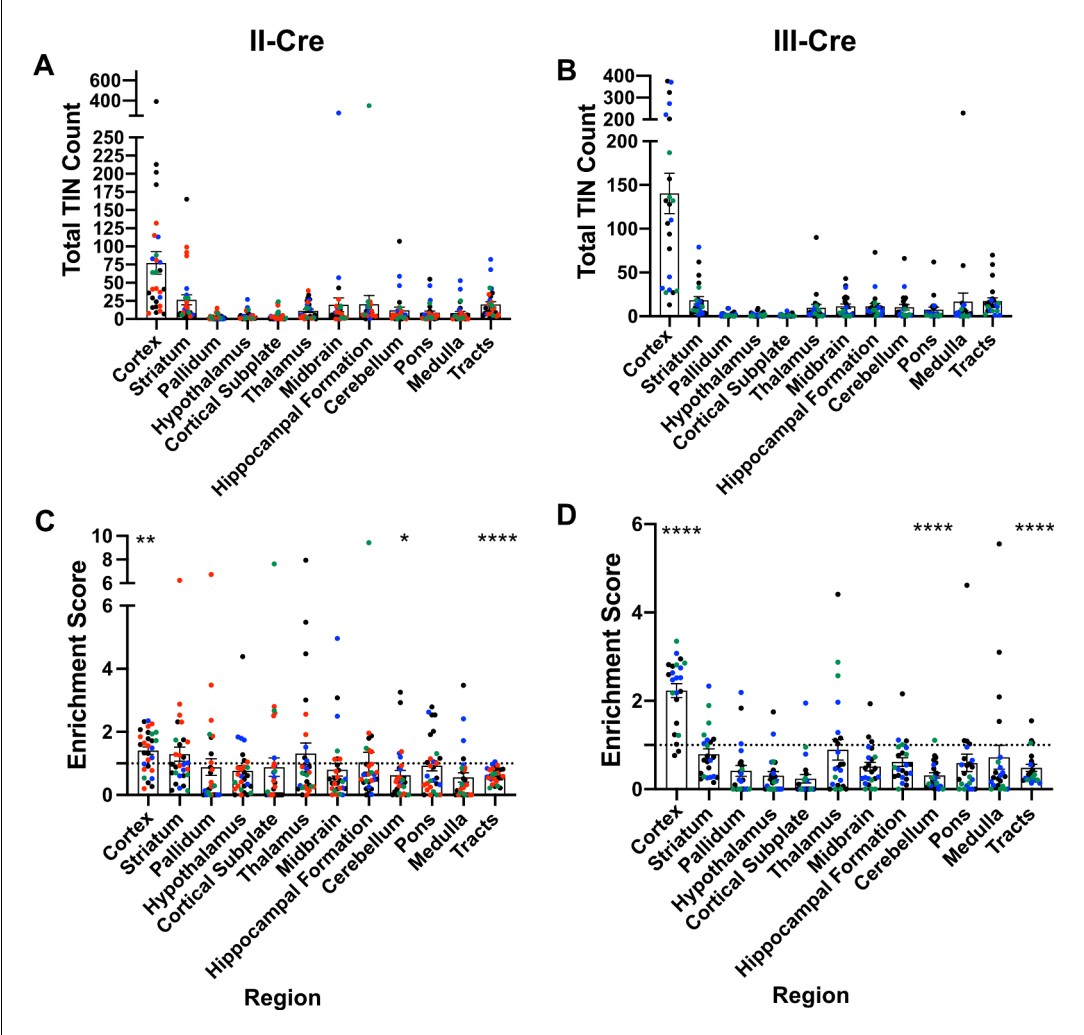

**Figure 1.** *Toxoplasma*-injected neurons (TINs) show a predilection for the cortex at 3 weeks post-infection. Cre-reporter mice were infected with II-Cre or III-Cre *Toxoplasma* parasites as indicated. Brains were harvested, sectioned, labeled, and quantified as previously described (*Mendez et al., 2018*). (A, B) Graphs of the absolute numbers of TINs mapped to 12 regions of the brain. (C, D) Graphs of TINs/region normalized to the size of the region. The dashed line is 1, the value at which the distribution of TINs would be considered proportional to the region size. Bars, mean ± SEM. N = 16–19/ sections/mouse. Individual colors denote animals from individual cohorts, N = 4–12 mice/cohort for II-Cre infected mice, 2–11 mice/cohort for III-Cre infected mice. (C, D) *p=0.0170, **p=0.0021, ****p≤0.0001, one-sample t-test. p-Values for all regions are in *Supplementary file 2*. Mice were excluded from analyses if GFP$^+$ cells were not above background rate of GFP$^+$ cells in saline-injected Cre reporter mice or if identified as an outlier by ROUT outlier test, which exclude both Cohort 1 (red) III-Cre infected mice. *Figure 1—figure supplement 1* includes all mice and uses all Allen Institute sections for normalization/enrichment index.

The online version of this article includes the following source data and figure supplement(s) for figure 1:

**Source data 1.** Raw data for TINs count for II-Cre and III-Cre cohort 1 (Red).
**Source data 2.** Raw data for TINs count for II-Cre and III-Cre cohort 2 (Green).
**Source data 3.** Raw data for TINs count for II-Cre and III-Cre cohort 3 (Blue).
**Source data 4.** Raw data for 10 mice each for TINs count for II-Cre and III-Cre cohort 4 (Black).
**Source data 5.** Raw data for remaining mice for TINs count for II-Cre and III-Cre cohort 4 (Black).
**Figure supplement 1.** Original data with all mice and using full Allen Atlas area.
**Figure supplement 2.** The visual, somatosensory, and motor cortices are highly enriched cortical regions containing *Toxoplasma*-injected neurons (TINs).

[GAD[+] interneurons make up approximately 15–20% of all cortical neurons (*Gentet et al., 2000*; *Lodato and Arlotta, 2015*)] as GAD staining is altered in *Toxoplasma*-infected brain (*Brooks et al., 2015*; *Carrillo et al., 2020*). Fortunately, parvalbumin labels 40–50% of GAD[+] interneurons and calbindin labels approximately 30% of GAD[+] interneurons (*Hof et al., 1999*; *Keller et al., 2019*). To quantify how often TINs were calbindin or parvalbumin interneurons, we labeled brain sections with antibodies against either calbindin or parvalbumin, and with antibodies against NeuN, a pan-neuronal marker. These labeled sections were then imaged with confocal microscopy to identify TINs that co-stained with NeuN and either calbindin or parvalbumin. Within the cortex, irrespective of infecting strain, approximately 5% (4.8 ± 2.0% II-Cre, 3.9 ± 0.6% III-Cre) of TINs were calbindin[+] (*Figure 2A,B*), while none were parvalbumin[+] (*Figure 2C,D*). In the striatum, we observed that approximately 4% of II-Cre TINs were calbindin[+] (4.4 ± 2.0) and approximately 9% of III-Cre TINs were calbindin[+] (9.4 ± 3.4%) (*Figure 2E,F*). As in the cortex, no TINs co-localized with parvalbumin[+]

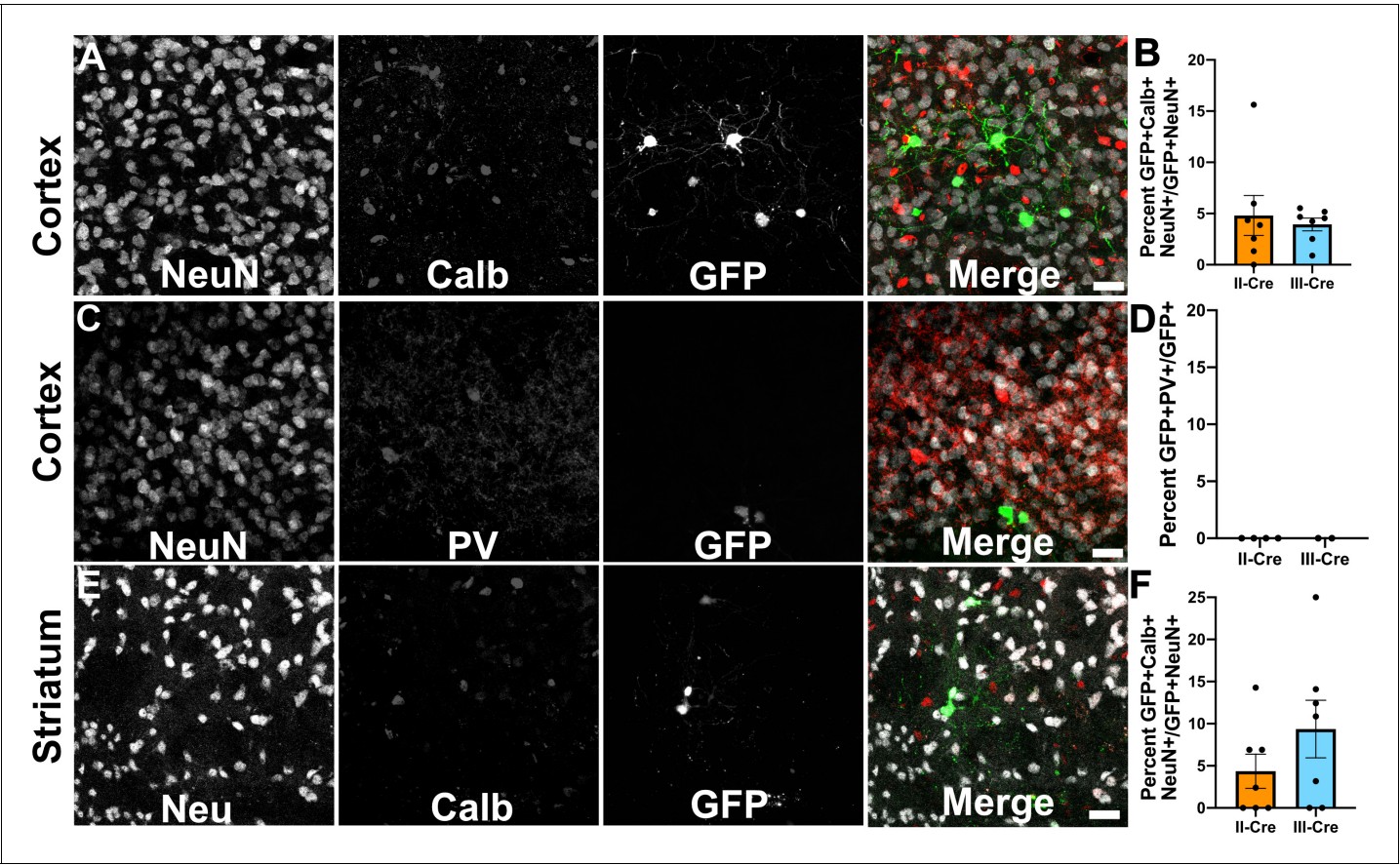

**Figure 2.** *Toxoplasma*-injected neurons (TINs) rarely co-localize with inhibitory interneurons. Forty micron brain sections from II-Cre or III-Cre infected mice were stained with anti-NeuN antibodies and either anti-calbindin or anti-parvalbumin antibodies. Stained sections were then imaged by confocal microscopy to determine co-localization between TINs (GFP[+]) and calbindin (Calb) or parvalbumin (PV) staining. (A) Representative images of a cortical region from a section stained for Calb. (B) Quantification of the percentage of TINs that co-localized with Calb staining. (C) As in (A) except the images are of a cortical region stained for PV. (D) As in (B) except for PV quantification. (E) As in (A), but in the striatum. (F) As in (B) except for striatal Calb quantification. For (A, C, E) Merge images, gray = NeuN, red = Calb or PV, and green = GFP. N = 9 sections/mouse, seven mice/group for Calb, 2–4 mice/group for PV. (B, D, F) Bars, mean ± SEM. (B) For II-Cre infected mice, 24–127 TINs/mouse were analyzed; for III-Cre infected mice, 254–503 TINs/mouse were analyzed. (D) For II-Cre, 8–68 TINs/mouse, and for III-Cre, 281, 346 TINs/mouse were analyzed. (F) For II-Cre infected mice, 3–290 TINs/mouse were analyzed; for III-Cre infected mice, 21–214 TINs/mouse were analyzed. No significant differences were identified between groups, Student's t-test.

The online version of this article includes the following source data for figure 2:

**Source data 1.** Raw numbers for quantification of Calb[+] cells.

**Source data 2.** Raw numbers for quantification of PV[+] cells.

neurons in the striatum (N = 6–65 TINs/mouse, four mice for II-Cre; N = 30 and 84 TINs/mouse, two mice for III-Cre).

These data suggest that *Toxoplasma* rarely interacts with or injects cortical or striatal interneurons, though we cannot exclude the possibility that *Toxoplasma*-injected interneurons die prior to the expression of GFP or that parasites primarily interact with interneurons that do not express parvalbumin or calbindin. Either scenario would limit our ability to detect these *Toxoplasma*-interneuron interactions. Within interneurons, *Toxoplasma* more commonly injects calbindin[+] interneurons, not parvalbumin[+] interneurons, an unexpected finding given the higher prevalence of parvalbumin[+] interneurons.

## TINs co-localize with FoxP2, a marker of cortical excitatory neurons and striatal MSNs

As only a small number of TINs co-localized with markers for inhibitory neurons, we next sought to determine if TINs co-localized with a marker for excitatory cortical neurons. For these studies, we selected the transcription factor FoxP2. FoxP2 is primarily expressed by glutamatergic/excitatory neurons within layer 6 of the mouse cortex, though some FoxP2[+] excitatory neurons are also found in cortical layers IV and V (*Arlotta et al., 2005*; *Lodato and Arlotta, 2015*). FoxP2 expression also occurs in medium spiny neurons (MSNs) in the striatum, allowing us to use a single marker for both cortical and striatal studies (*Ferland et al., 2003*; *Lai et al., 2003*). We quantified how often TINs were FoxP2[+] by labeling and imaging brain sections as above except now using antibodies directed against FoxP2. Within the cortex, irrespective of infecting *Toxoplasma* strain, approximately 58% (58.0 ± 3.8% II-Cre, 58.3 ± 2.3% III-Cre) of the analyzed TINs showed FoxP2 co-localization (*Figure 3A,B*). In the striatum, we found that approximately 68% (68.1 ± 7.2%) of analyzed II-Cre TINs and approximately 83% (83 ± 4.1%) of analyzed III-Cre TINs showed FoxP2 co-localization (*Figure 3C,D*).

This high rate of co-localization between striatal TINs and FoxP2 suggested that striatal TINs were MSNs, which would be expected given that 90–95% of neurons in the striatum are MSNs (*Ouimet et al., 1984*; *Ouimet and Greengard, 1990*). As the identification of striatal TINs as MSNs had important implications for pursuing electrophysiology studies, we sought to further confirm that striatal TINs were MSNs by determining the rate of co-localization between striatal TINs and DARPP32, another marker for MSNs (*Ouimet et al., 1984*; *Ouimet and Greengard, 1990*). We found that approximately 20% (19.9 ± 5.7%) of II-Cre striatal TINs and approximately 25% (23.6 ± 4.8%) of III-Cre striatal TINs co-localized with DARPP32[+] staining (*Figure 3E,F*).

Given the relatively low percentage of TINs co-localizing with DARPP32 and prior work suggesting *Toxoplasma* infection alters protein localization and expression (*Cekanaviciute et al., 2014*; *Brooks et al., 2015*; *David et al., 2016*), we sought to determine how striatal FoxP2 and DARPP32 staining differed between uninfected and infected mice, and if this staining differed by proximity to TINs. To determine the abundance of FoxP2 and DARPP32 neurons near TINs, we re-analyzed the images used for *Figure 3*, this time quantifying the number of GFP[-] neurons (NeuN[+] cells) that co-localized with FoxP2 or DARPP32. To determine the abundance of FoxP2 and DARPP32 neurons regardless of proximity to TINs, we generated new, randomly distributed images from a subset of stained sections used for *Figure 3* and quantified the co-localization between GFP[-] neurons and FoxP2 or DARPP32. We found that the average number of FoxP2[+] neurons in each image did not differ between images from either uninfected or infected mice, regardless of infecting *Toxoplasma* strain (*Figure 3—figure supplement 1A*); nor did the abundance vary in proximity to TINs (*Figure 3—figure supplement 1B*). The average number of DARPP32[+] neurons in each image also did not differ between random images from uninfected or infected mice (*Figure 3—figure supplement 1C*). Conversely, in images taken in proximity to TINs, the number of DARPP32[+] neurons showed a statistically significant decrease (*Figure 3—figure supplement 1D*). In addition, we noted that, on average, more striatal neurons were FoxP2[+] than DARPP32[+], even in uninfected mice (*Figure 3—figure supplement 1A,C*). To further test the possibility that FoxP2 identifies more striatal neurons than DARPP32, we stained tissues sections for both FoxP2 and DARPP32 and compared the co-localization between these proteins. Consistent with the findings in the single stains, only approximately 70% of FoxP2[+] neurons stained for DARPP32 (69.6 ± 0.3% saline, 68.6 ± 0.6% II-Cre, 73.0 ± 4.0% III-Cre) (*Figure 3—figure supplement 2A*), while 99% (99.7 ± 0.0% saline, 99.8 ± 0.0% II-Cre, 99.8 ± 0.1% III-Cre) of DARPP32[+] neurons stained for FoxP2 (*Figure 3—figure supplement 2B*).

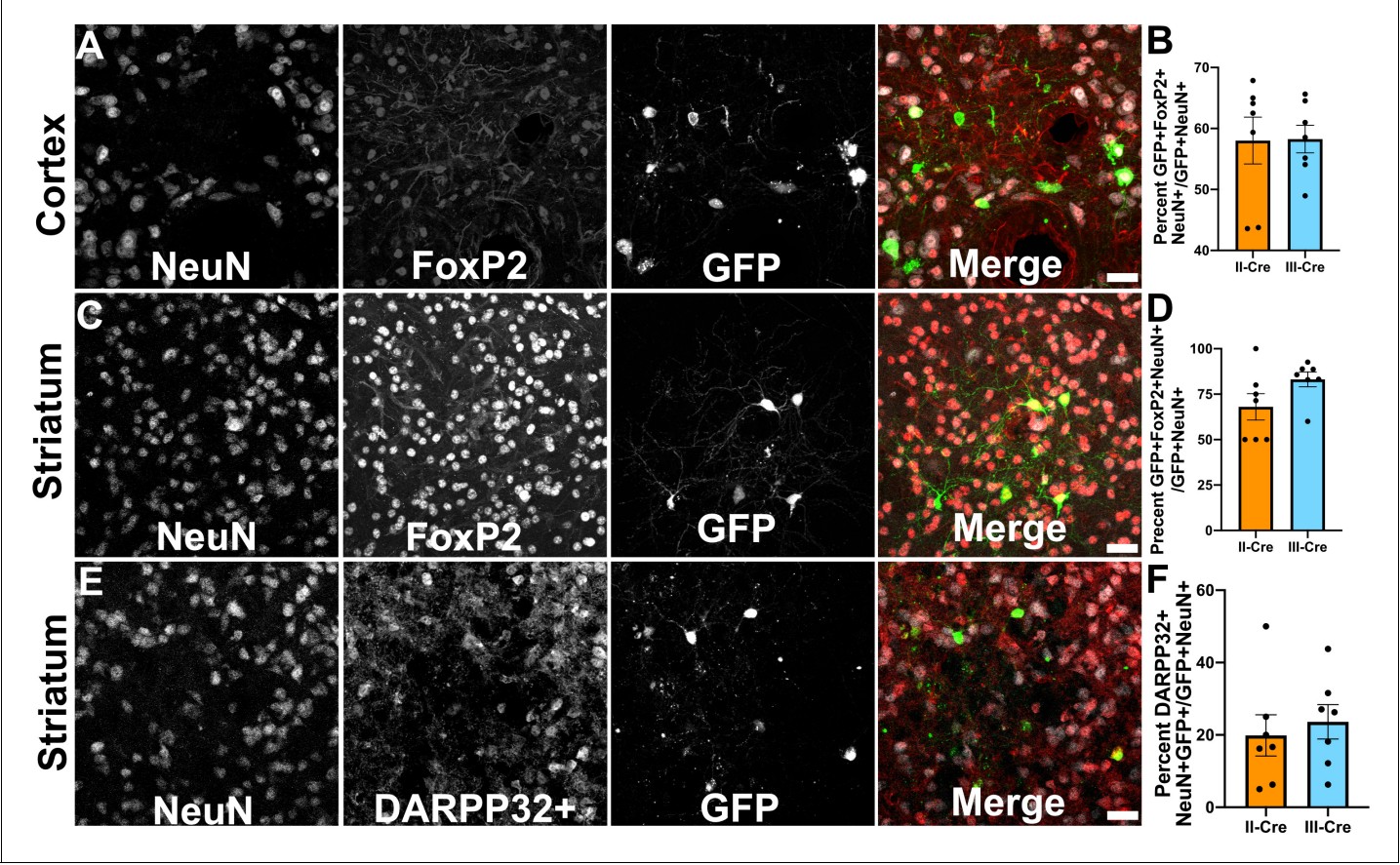

**Figure 3.** *Toxoplasma*-injected neurons (TINs) co-localize with FoxP2 and DARPP32 staining. Brain sections from II-Cre or III-Cre infected mice were stained with anti-NeuN antibodies and anti-FoxP2 or anti-DARPP32 antibodies. DARPP32 staining was only done in the striatum. Stained sections were then analyzed by confocal microscopy to determine co-localization between TINs (GFP[+]) and FoxP2 or DARPP32 staining. (A) Representative image of a cortical region from a section stained as labeled. (B) Quantification of the percentage of TINs that co-localized with FoxP2 staining. (C) As in (A) except the images are of a striatal region. (D) As in (B) except for striatal TINs. (E) As in (C) except the tissue is stained with anti-DARPP32 antibodies (as labeled). (F) As in (D) except co-localization is between TINs and DARPP32 staining. (A, C, E) Merge images, gray = NeuN, red = FoxP2 or DARPP32, and green = GFP. Scale bar = 50 µm. (B, D, F) Bars, mean ± SEM. For (B, D, F) N = 9 sections/mouse, seven mice/group. (B) For II-Cre infected mice, a total of 24–127 TINs/mouse were analyzed; for III-Cre infected mice, 254–503 TINs/mouse were analyzed. (D) For II-Cre infected mice, a total of 28–68 TINs/mouse were analyzed; for III-Cre infected mice, 290–858 TINs/mouse were analyzed. (F) For II-Cre infected mice, 6–237 TINs/mouse were analyzed; for III-Cre infected mice, 16–198 TINs/mouse were analyzed. No significant differences were identified between groups, Student's t-test.

The online version of this article includes the following source data and figure supplement(s) for figure 3:

**Source data 1.** Raw numbers for quantification of DARPP32[+] cells.

**Source data 2.** Raw numbers for quantification of FoxP2[+] cells.

**Figure supplement 1.** Quantification of FoxP2[+] or DARPP32[+]neurons in the striatum.

**Figure supplement 2.** Baseline quantification for DARPP32[+] and FoxP2[+]co-localization in the striatum.

These data suggest that striatal TINs are likely MSNs as they co-localize with markers of MSNs, especially FoxP2. The higher co-localization of TINs with FoxP2 versus DARPP32 is likely driven by the higher number of FoxP2[+] neurons within the striatum as well as the sensitivity of DARPP32 expression/staining to disruption by infection or inflammation. Given that FoxP2 stains more striatal neurons than DARPP32, even in uninfected mice, FoxP2 may be a more sensitive marker for striatal MSNs than DARPP32.

## Passive electrophysiologic properties of bystander MSNs are relatively stable

The identification of striatal TINs as co-localizing with markers for MSNs offered us an unusual opportunity. MSNs make up 90–95% of dorsal striatal neurons (*Mensah and Deadwyler, 1974*;

*Graveland and DiFiglia, 1985*; *Gerfen, 1992*) and are highly characterized, including at the level of single cell electrophysiology in situ (i.e., patch clamping in ex vivo slices). Targeting such a well-characterized, high-frequency population would allow us to directly compare the electrophysiology of TINs and bystander neurons—striatal neurons in proximity to a TIN but not injected with *Toxoplasma* protein—as both groups of neurons would likely be MSNs. We reasoned that comparing the physiology of TINs and bystanders would allow us to determine the role of general inflammation in driving any changes we observed in TINs (i.e., if TINs and bystanders showed similar physiologic changes, then these changes are likely driven by the general inflammatory response to *Toxoplasma*, not from direct manipulation of TINs by *Toxoplasma* effector proteins). For these studies, we chose to record from III-Cre infected mice because the type III strain produces an higher frequency of TINs than the type II strain (*Figure 1A*), and because III-Cre TINs show higher rates of co-localization with the MSN marker FoxP2 (*Figure 3F*).

To ensure our capability to patch-clamp onto MSNs in ex vivo slices, we obtained thick brain slices from uninfected mice and performed patch clamping using a standard protocol (*Nisenbaum et al., 1994*; *Suter et al., 1999*; *Haubensak et al., 2010*; *Wang et al., 2019*). As expected, in uninfected mice, dorsal striatum neurons showed classical MSN electrophysiologic properties such as a hyperpolarized resting membrane potential of approximately −80 mV and a delayed first action potential (AP) (*Figure 4A*). In addition, when we filled a neuron showing the above characteristics and stained the section for DARPP32, we found the filled neuron co-localized with DARPP32 labeling (*Figure 4B–D*).

In infected mice, we electrophysiologically interrogated bystander MSNs and TINs, with bystander MSNs being approximately 30–100 μm from a TIN. Bystander MSNs had a resting membrane potential of −69.0 ± 1.2 mV, which is mildly depolarized compared to MSNs in uninfected mice (*Figure 5A*, *Figure 5—figure supplement 1*). Consistent with a depolarized resting membrane potential, bystander MSNs required fewer steps of injected current to reach the threshold to fire the first AP compared to MSNs in uninfected mice (12 ± 0.3 steps for bystanders and 13 ± 0.4 for MSNs in uninfected mice) (*Figure 5—figure supplement 2*). Besides these two properties, the following electrophysiologic properties were equivalent when comparing bystander MSNs to MSNs from uninfected mice: AP threshold, after hyperpolarization peak, AP amplitude, delay to first AP, and input resistance (*Figure 5B–G*).

These results indicate that though there is a mild change in the resting membrane potential of bystander MSNs during infection, the infected/inflammatory microenvironment at this time point causes relatively little change to neurons that have not directly interacted with *Toxoplasma*.

## TINs show a highly depolarized membrane potential

We next evaluated TINs by following the same procedure as when we recorded from uninfected and bystander MSNs, except that we used the expression of GFP to identify the TIN (*Figure 6—figure supplement 1*). Unexpectedly, we were unable to properly patch onto most striatal TINs (i.e., we were unable to form a gigaseal). We were able to record from a total of 10 TINs, most of which had highly depolarized resting membrane potentials (−49.1 ± 4.5 mV) compared to either bystander or uninfected MSNs (*Figure 6A*). None of these TINs generated APs

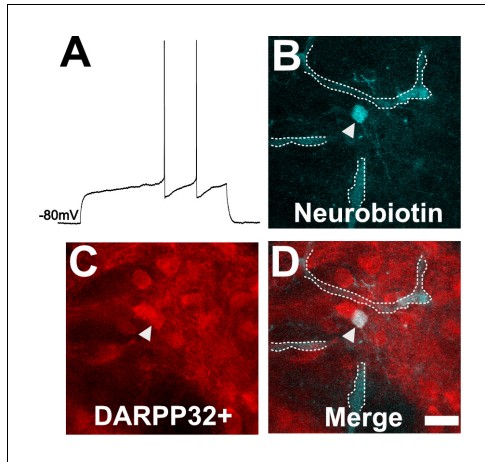

**Figure 4.** Neuron with electorphysiology of a medium spiny neuron (MSN) co-localizes with DARPP32 staining. (A) Sample tracing from the shown neuron. Note the hyperpolarized resting membrane potential of −80 mV and the delayed time to the first action potential. (B) Image of patched neuron filled with neurobiotin. Arrowhead points to filled neuron. Dashed white line denotes biotin-filled blood vessels. (C) Image of DARPP32 staining. After recording and filling, the section was fixed and counter-labeled with anti-DARPP32 antibodies. (D) Merge of images **B** and **C**. (B–D) Section imaged on a Zeiss 880 confocal microscope. The shown images are a maximum projection of 12, 1 μm step images, from a 100 μm z-stack. Scale bar = 50 μm.

in response to the standard protocol, possibly because this highly depolarized membrane potential is too close to the threshold for firing an AP (~45 mV) in MSNs (*Figure 5B*). Several possibilities could most easily explain this highly depolarized membraned potential: (i) the patched GFP$^+$ cells were not neurons, (ii) the patched cells were dead neurons, or (iii) the patched cells were unhealthy neurons. To distinguish between these possibilities, when possible, we hyperpolarized the patched TINs, followed by attempts to generate APs. For five TINs, after we applied between −20 and −180 pA of negative current, membrane potentials of −55 to −66 mV were observed. With this 'normalization' of the resting membrane potential, we found that TINs could generate APs (*Figure 6B*), definitively showing that they were neurons. Finally, to ensure the depolarized resting membrane potential of TINs was not due to fluorescent protein expression in the setting of inflammation, we recorded from YFP$^+$ neurons in infected mice that constitutively express YFP in a subset of cortical neurons (*Feng et al., 2000*). In these mice, we found a range of resting membrane potentials that is typical of cortical neurons (*Pangratz-Fuehrer and Hestrin, 2011*; *Oswald et al., 2013*) and distinct from the physiologic abnormalities observed in TINs (*Figure 6—figure supplement 2*).

Given that such findings are consistent with the electrophysiology of sick or dying neurons (*Lipton, 1999*), we compared the total number of TINs in brain sections from mice infected with III-Cre parasites for 3 or 8 weeks. Consistent with the possibility that the electrophysiology reflects that TINs are sick/dying, brain sections from 8 wpi mice showed an approximately 90% reduction in the number of TINs compared to brain sections from three wpi mice (*Figure 7*).

Collectively, our data suggest that neurons that directly interact with *Toxoplasma* have highly aberrant electrophysiology and most will die over a relatively short period. On the other hand, bystander neurons, which are in a similar inflammatory environment, show relatively minor changes in terms of electrophysiology. Together, these data suggest that neuron pathology is extremely targeted, even in a highly inflammatory setting.

## Discussion

In this study, we leveraged the *Toxoplasma*-Cre system to extend our understanding of neuron-*Toxoplasma* interactions. Across multiple cohorts of mice and using two genetically distinct *Toxoplasma* strains, our data show that TINs consistently localize to the cortex and striatum. For the cortex, the number of TINs is higher than would be expected even after accounting for the size of the cortex. Within the cortex and striatum, we found little evidence of *Toxoplasma* targeting a specific neuron subtype, as TINs co-localize with markers for highly abundant neurons in a region (e.g., MSNs in the striatum). Interestingly, when we used whole cell patch clamping in ex vivo slices to assess both striatal TINs and bystander MSNs, we found discrepant electrophysiology. Bystander MSNs showed mild changes in resting membrane potential and no changes in other passive firing properties, while TINs had highly depolarized resting membrane potentials and did not fire APs unless artificially hyperpolarized. These data suggest that TINs, but not bystanders, are unhealthy and potentially dying, a possibility confirmed by a drastic decrease in the number of TINs by 8 wpi. Together, these data offer an unprecedented look into the neuron-*Toxoplasma* interface, raising new questions about why the cortex is particularly vulnerable to *Toxoplasma* infection and the evolutionary advantage of the death of neurons injected with *Toxoplasma* protein.

One of the major advances of this work is how we mapped neuron-*Toxoplasma* interactions. Most previous studies have utilized a single *Toxoplasma* strain, relatively few animals (or more animals but relatively few sections/animal), and were limited to measuring cyst location. On the other hand, we mapped TINs, which outnumber cysts (*Koshy et al., 2012*) and show uniform distribution of GFP, allowing us to map neuron cell bodies/somas which is the traditional method for identifying neuron anatomic location. Furthermore, as our semi-automated method is relatively quick (*Mendez et al., 2018*), in total, we mapped a total of over 5000 TINs/*Toxoplasma* strain, using multiple cohorts of mice and >15 sections/mouse. These robust methods offer confidence that our findings are not unduly influenced by a single outlier mouse, a cohort effect, or insufficient sampling for a process that is highly heterogenous. Our findings are consistent with several aspects of the two most extensive cyst mapping studies (*Berenreiterová et al., 2011*; *Boillat et al., 2020*). All three studies found: (i) high levels of inter-mouse variability; (ii) identified the cortex—and especially the motor, somatosensory, visual cortical regions—as showing enrichment for *Toxoplasma* presence (TINs or cysts) even when accounting for region size; and (iii) determined that the cerebellum shows

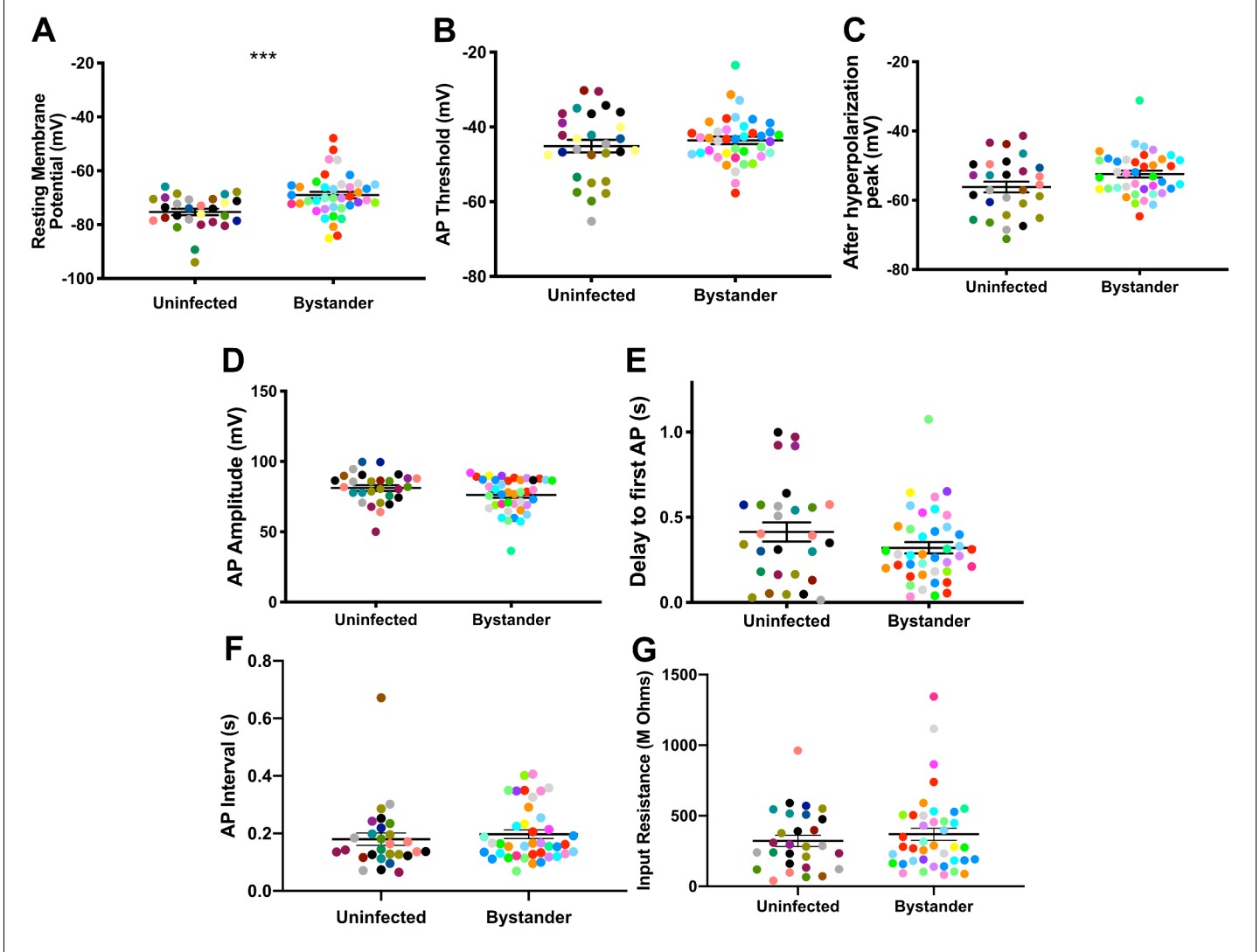

**Figure 5.** Bystander medium spiny neurons (MSNs) show similar electrophysiology as MSNs from uninfected mice. (**A**) Graph of the resting membrane potential of whole cell patch clamped MSNs in uninfected mice (−74.9 ± 2.1 mV) and bystander MSNs in infected mice (−68.7 ± 4.0 mV). (**B**) Action potential (AP) threshold, (**C**) after hyperpolarization peak, (**D**) AP amplitude, (**E**) delay to first AP, (**F**) AP interval, and (**G**) input resistance in MSNs from uninfected mice and bystander MSNs in infected mice. Dots represent individual MSNs. Matching color dots denote cells from the same mouse. Uninfected MSNs, N = 28 cells recorded from 14 mice, 1–5 cells recorded/mouse. Bystander MSNs, N = 40 cells recorded from 18 mice, 1–5 cells recorded/mouse. Line, mean ± SEM. ***p=0.0007, Mann-Whitney U-test.

The online version of this article includes the following source data and figure supplement(s) for figure 5:

**Source data 1.** Raw electrophysiology recording data for MSNs from uninfected mice and bystander MSNs and TINs from infected mice (source data for *Figures 5* and *6A*).

**Figure supplement 1.** Schematic of bystander and *Toxoplasma*-injected neurons (TINs) during recording procedure.

**Figure supplement 2.** Bystander medium spiny neurons (MSNs) fire the first action potential (AP) sooner than MSNs from uninfected controls.

**Figure supplement 2—source data 1.** Raw data for the number of steps quantified to reach first action potential.

relatively little *Toxoplasma* presence, especially when accounting for size. These consistencies suggest that regional cyst burden is driven by differences in the number of neuron-parasite interactions, not by differences in regional innate immune responses as has been suggested for West Nile virus (WNV) infection (*Cho et al., 2013*). In addition, these regional differences are consistent with the largest studies in AIDS patients with toxoplasmic encephalitis, in which more lesions are identified in the cortex and the basal ganglia compared to the cerebellum (*Porter and Sande, 1992*; *Arendt et al., 1999*), highlighting the relevance of the mouse model to human disease. Currently,

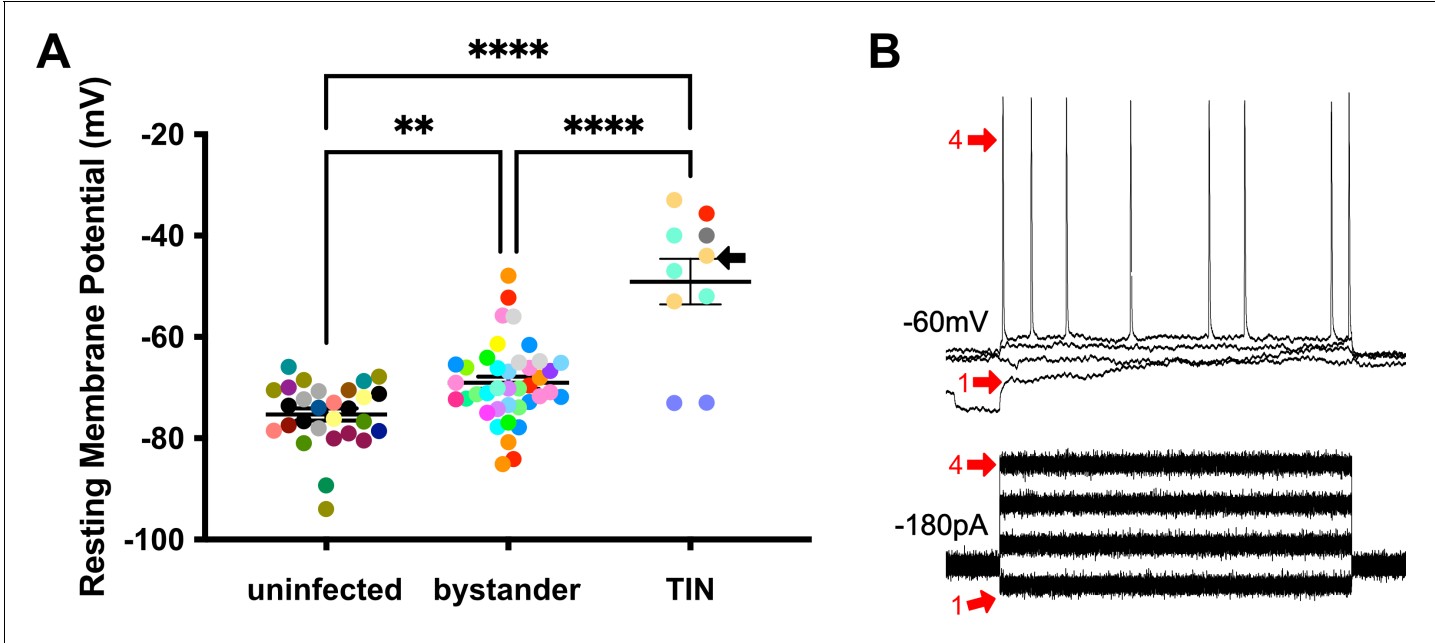

**Figure 6.** Unlike bystander neurons, *Toxoplasma*-injected neurons (TINs) have highly abnormal electrophysiology. (**A**) Graph of the resting membrane potential of whole cell patch clamped medium spiny neurons (MSNs) in uninfected mice (−74.9 ± 2.1 mV) and bystander MSNs (−68.7 ± 4.0 mV) and TINs (−49.1 ± 4.5 mV) in infected mice. **p=0.0086, ****p=0.001, one-way ANOVA with Tukey's post-test. For TINs, N = 10 cells recorded from six mice; uninfected and bystander described in *Figure 5*. Black arrow identifies TIN shown in (**B**). Lines, mean ± SEM. (**B**) *Top*: Example traces from TIN after hyperpolarization. *Bottom*: Visualization of a portion of the current injection protocol. The identified TIN was hyperpolarized to a resting membrane potential of −60 mV by the injection of −180 pA of current. The red arrow denotes a single hyperpolarized step. The numbers (1, 4) match the voltage measurement (tracings) with the steps of increasing current. Note that an action potential is finally generated at the fourth step of increasing current.

The online version of this article includes the following figure supplement(s) for figure 6:

**Figure supplement 1.** Representative images of *Toxoplasma*-injected neurons (TINs) used for recording.

**Figure supplement 2.** YFP[+] cortical neurons from infected mice show typical resting membrane potentials.

we do not know what drives these regional differences in tropism, though one possibility is differences in the transmissibility of *Toxoplasma* across the vasculature that supplies the cortex and striatum/basal ganglia versus the cerebellum (anterior versus posterior circulation, respectively), a possibility also suggested for WNV (*Daniels et al., 2017*).

A second advance in our study is the ability to identify the type of neuron injected by *Toxoplasma*. The finding that TINs are comprised of common neuron subtypes in the deep cortex and striatum (e.g., FoxP2[+] neurons) suggests that *Toxoplasma* injects (and presumptively infects) in proportion to the neuron subtypes in a given region. One exception to this rule is the lack of TINs that also express parvalbumin. Parvalbumin-expressing neurons make up approximately 40–50% of the cortical GABAergic (inhibitory) neurons and are found in all deeper layers of the cortex, while calbindin-expressing neurons are less frequent, especially in layers IV-VI of the cortex (*Hof et al., 1999*; *Tremblay et al., 2016*). One possibility for the lack of parvalbumin[+] TINs is a decrease or change in parvalbumin expression/location as seen for DARPP-32 (*Figure 3—figure supplement 1*) or GAD-67 (*Brooks et al., 2015*). Another more exciting possibility is that parvalbumin[+] neurons have fewer interactions with *Toxoplasma* because they have a high concentration of perineural nets—extracellular matrix that surrounds different parts of the neuron (*Baker et al., 2017*)—blocking *Toxoplasma*'s ability to contact parvalbumin[+] neurons. Another possibility is that parvalbumin[+] neurons are highly sensitive to injection with *Toxoplasma* protein and rapidly die before expressing GFP and/or die prior to the time point at which we performed our studies. Future studies will define the factors that determine which neurons interact with parasites.

A final advance in this work is the electrophysiology comparing TINs and bystander neurons. To the best of our knowledge, these studies are the first in any neurotropic infection to compare the in

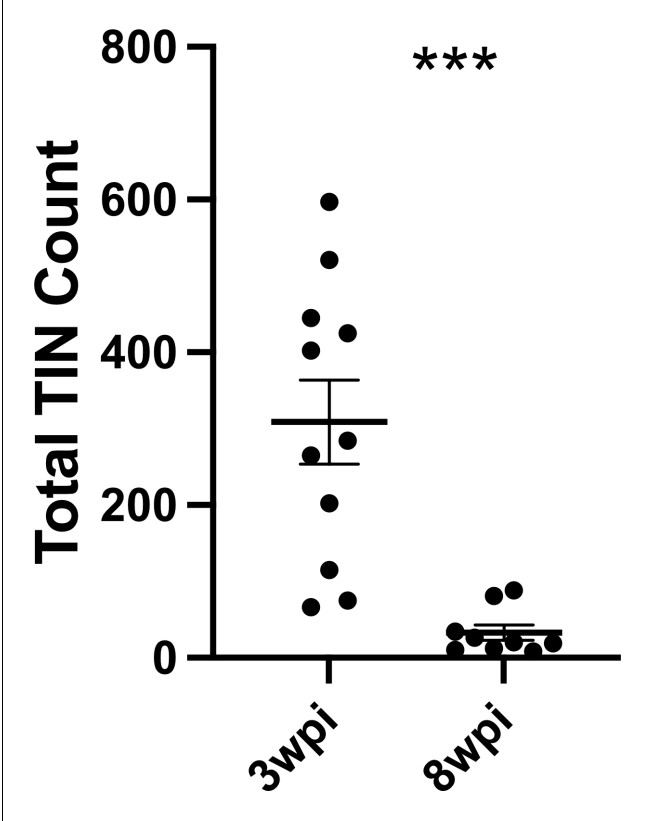

**Figure 7.** Number of *Toxoplasma*-injected neurons (TINs) decrease by 8 weeks post-infection (wpi). Mice were infected and, at 3 or 8 wpi, analyzed as in *Figure 1* except that total TINs numbers were quantified. Lines, mean ± SEM. ***p≤0.0001, Mann-Whitney U-test; 16–19 sections/mouse, N = 9–11 mice/time point. The online version of this article includes the following source data for figure 7:

**Source data 1.** Raw data for quantification of TINs count at 3 and 8 wpi.

situ electrophysiology between microbe-interacted (or infected) neurons versus those neurons in close proximity but without direct microbial interactions. The utility of comparing TINs and bystanders is to address what physiologic effects are driven by the general neuroinflammatory response versus those changes driven by neuron-parasite interactions. Given the relatively common understanding that microglia will strip synapses off infected and uninfected neurons in the inflamed brain (*Vasek et al., 2016*; *Di Liberto et al., 2018*), we found it remarkable that bystander MSNs physiology showed only minor changes compared to MSNs from uninfected mice while TINs showed highly abnormal electrophysiology (*Figure 6*). One possible explanation for the minimal changes seen in bystanders is that these neurons undergo compensatory mechanisms that allow them to avoid drastically changing, as a dramatic response would be detrimental to the local circuitry. Another possibility is that the effect of the neuroinflammatory response is confined to a small area near TINs, leaving these more distant bystanders relatively spared. Either explanation would preserve circuit function at a global level and allow for a relatively quick reversal of these physiologic changes once the inflammatory response subsides. Of note, while the physiology of the bystanders is relatively normal compared to TINs, this depolarized resting membrane potential is still abnormal and the decrease in required input current (*Figure 6—figure supplement 2*) suggests these neurons are mildly hyperexcitable. If similar changes also occur in other bystander neurons (e.g., cortical neurons) and in other CNS infections, they might explain why patients with active brain infections have an increased likelihood of seizing (*Luft and Remington, 1992*; *Neu et al., 2015*). These mild abnormalities might also summate across many bystander neurons to cause circuit level changes. While we did not test this possibility, such changes in the far more numerous bystanders—especially if the changes are reversible—could explain both the heterogeneity of behavioral changes seen in infected

animals (*Worth et al., 2014*) and the relative lack of behavioral abnormalities observed in *Toxoplasma*-infected mice that have cleared both the infection and the immune response (*McGovern et al., 2020*).

TINs—which include actively infected neurons and uninfected, injected neurons (*Figure 5—figure supplement 1*)—show highly abnormal electrophysiology, only firing APs after artificial hyperpolarization. While we cannot distinguish between whether this abnormal physiology leads to neuron death or the sick state of these neurons drives the abnormal physiology (we favor the latter possibility), the discrepancy between the physiology of TINs and bystanders suggests that the root cause is specific to neurons injected with *Toxoplasma* protein. Such restricted changes could arise from direct manipulation of TINs by the injected parasite proteins or selective microglial/macrophage recognition and phagocytosis of TINs, consistent with what was observed in lymphocytic choriomeningitis virus infection (*Di Liberto et al., 2018*). Another possibility, as noted above, is that the immune response is tightly regulated and confined, such that the destruction leveled by this response is limited to a very small area around TINs. Such a possibility is consistent with prior work showing that partial abrogation of astrocytic TGF-β signaling led to de-regulation of the CNS immune response to *Toxoplasma* and higher levels of neuronal loss without changing the CNS parasite burden (*Cekanaviciute et al., 2014*). Future studies will determine what factors drive this abnormal physiology.

Finally, our finding of an approximately 90% decrease in TINs by 8 wpi suggests that we need to re-evaluate our understanding of how and why microbes persist in or even infect neurons. Microbes with the capability to infect multiple cell types are commonly found in neurons, not astrocytes (e.g., *Toxoplasma*, WNV, measles virus). This neuronal preference has been explained by the idea that—given the long-lived, relatively non-regenerative nature of neurons—neuron survival must be ensured at almost any cost. Thus, rather than initiating an immune response that might result in neuron death, neurons survive infection through non-cytolytic clearance of microbes (*Binder and Griffin, 2001*; *Patterson et al., 2002*; *Samuel and Diamond, 2005*; *Miller et al., 2016*) and/or pushing the infecting microbes to enter a latent state (*Speck and Simmons, 1991*; *Ferguson and Hutchison, 1987*; *Tanaka et al., 2016*). The death of TINs directly challenges this model, suggesting that the battle to persist in neurons is far more nuanced than previously recognized. For example, given that >90% of TINs are expected to be uninfected (*Koshy et al., 2012*), we would predict that many uninfected neurons are dying. What is the advantage of uninfected, parasite-manipulated neurons dying? One possibility is that these TINs are so abnormal (from either direct manipulation by parasites or because of neuron-immune cell interactions) that global brain function is better preserved with the removal of these neurons. Such a possibility is consistent with the altered gene expression identified in neurons that cleared an attenuated rabies virus infection (*Gomme et al., 2012*). Another possibility is that the death of TINs—both infected and uninfected—is essential for parasite control and perhaps the death of uninfected TINs is simply collateral damage. Or the death of uninfected TINs might divert immune attention from the rare, persistently infected neuron. Regardless of whether this death benefits the host or microbe (or both), based upon the data presented here and previously, we offer an alternative model for neuron-tropic microbes: neurons mount appropriate immune responses to both cytokines and infections such that persistent infection is a rare outcome, rather than a universal one. Limited data in a murine model of measles virus infection is consistent with this possibility (*Miller et al., 2019*), though future studies using other neurotropic pathogens will be required to definitively determine how universal this model is.

## Materials and methods

### Key resources table

| Reagent type (species) or resource | Designation | Source or reference | Identifiers | Additional information |
|---|---|---|---|---|
| Strain, strain background (*Toxoplasma* gondii) | Type II (Prugniaud) | *Koshy et al., 2012* | II-Cre | Expresses mCherry and Cre recombinase |

*Continued on next page*

*Continued*

| Reagent type (species) or resource | Designation | Source or reference | Identifiers | Additional information |
|---|---|---|---|---|
| Strain, strain background (*Toxoplasma* gondii) | Type III (CEP) | *Christian et al., 2014* | III-Cre | Expresses mCherry and Cre recombinase |
| Strain, strain background (*Mus musculus*) | Ai6 mouse, ZsGreen1 | Jackson Laboratories | Stock # 007906 RRID:IMSR_JAX:007906 | |
| Strain, strain background (*Mus musculus*) | thy1-YFP-H mouse | Jackson Laboratories | Stock # 003782 RRID:IMSR_JAX:003782 | |
| Antibody | Anti-ZsGreen, (rabbit polyclonal) | Clontech | Cat# 632474 RRID:AB_2491179 | IHC (1:10,000) |
| Antibody | Anti-rabbit (goat polyclonal biotinylated conjugated) | Vector Labs | Cat# BA-1000 RRID:AB_2313606 | IHC (1:500) |
| Antibody | Anti-NeuN B-clone A60 (mouse biotin conjugated) | Millipore | Cat# MAB377 RRID:AB_177621 | IF (1:500) |
| Antibody | Anti-calbindin, (rabbit polyclonal) | Sigma-Aldrich | Cat# C2724 RRID:AB_258818 | IF (1:500) |
| Antibody | Anti-parvalbumin, (rabbit polyclonal) | Abcam | Cat# ab11427 RRID:AB_298032 | IF (1:1000) |
| Antibody | Anti-FoxP2, (rabbit polyclonal) | Abcam | Cat# ab16046 RID:AB_2107107 | IF (1:1000) |
| Antibody | Anti-DARPP32, (rabbit polyclonal) | Abcam | Cat# ab40801 RRID:AB_731843 | IF (1:500) |
| Antibody | Anti-rabbit IgG Alexa Fluor 568 (goat polyclonal) | Invitrogen | Cat# A11011 RRID:AB_143157 | IF (1:500) |
| Antibody | Cy5 streptavidin | Invitrogen | Cat# SA1011 | IF (1:500) |
| Commercial Assay or kit | Avidin-biotin complex kit | Thermo Fisher Scientific | Cat# 32020 | |
| Commercial Assay or Kit | 3,3'-Diaminobenzidine (DAB) Vectastain | Vector Labs | SK-4100 RRID:AB_2336382 | |
| Other | Neurobiotin | Vector Labs | SP1120 RRID:AB_2336606 | 0.6% |
| Other | DAPI stain | Invitrogen | D1306 RRID:AB_2629482 | (1 μg/ml) |
| Software | pCLAMP | Molecular Devices | v10.7 RRID:SCR_011323 | |
| Software | Prism | GraphPad | v9.1.0 RRID:SCR_002798 | |
| Software | MATLAB | MathWorks | v2015a | |

## Parasite maintenance

Parasites were maintained via serial passage through human foreskin fibroblasts using DMEM supplemented with 2 mM glutagro, 10% fetal bovine serum, and 100 IU/ml penicillin/100 μg/ml streptomycin. The type II (Prugniaud) and type III (CEP) strains used have been engineered to express mCherry and Cre recombinase and have been previously described (*Koshy et al., 2012*; *Christian et al., 2014*).

## Mice

Most mice used in these studies are Cre-reporter mice that express GFP only after the cells have undergone Cre-mediated recombination (*Madisen et al., 2010*). For *Figure 6—figure supplement 2* (recordings of YFP$^+$ cortical neurons), thy1-YFP-H mice were used (*Feng et al., 2000*). Mice were originally purchased from Jackson Laboratories (stock # 007906 for Cre-reporter mice, stock # 003782 for thy1-YFP mice) and bred in the University of Arizona Animal Care Facilities.

## Infections

Mice were infected at 2–3 months of age via intraperitoneal injection with freshly syringe-released parasites for *Figures 1*, *2*, *3* and *7*, and related figure supplements. Mice were inoculated with 10,000 freshly syringe-released parasites in 200 µl of USP-grade PBS for both II-Cre and III-Cre strains. To increase the reliability of electrophysiology studies (*Figures 5* and *6*, and related *Figure 6—figure supplements 1* and *2*), mice were infected at 5–6 weeks of age with III-Cre parasites only.

## Tissue preparation

For the localization studies, at 3 wpi animals were sedated with a ketamine/xylazine cocktail, intracardially perfused with saline followed by 4% paraformaldehyde (PFA), after which brains were harvested. Sections were then prepared as previously described (*Mendez et al., 2018*). In brief, left and right brain hemispheres were isolated, drop-fixed in 4% PFA followed by cryopreservation in 30% sucrose. Forty-micron-thick sagittal sections were generated using a freezing sliding microtome (Microm HM 430). Sections were then stored as free-floating sections in cryoprotective media (0.05 M sodium phosphate buffer containing 30% glycerol and 30% ethylene glycol) until labeling procedure was to be done. Sections for localization studies (*Figure 1*, *Figure 1—figure supplements 1* and *2*) were sampled every 200 µm, while for co-localization studies (*Figures 2* and *3*, *Figure 3—figure supplements 1* and *2*), sections were sampled every 400 µm.

## Immunohistochemistry

To ensure adhesion of tissue onto slides for localization studies, tissue was air-dried onto slides overnight then heated on a slide warmer for 40 min at 34˚C. Then tissue was dehydrated using increasing then decreasing concentrations of 50%, 75%, 95%, and 100% ethanol. Slides were washed with TBS, peroxidase inactivated (3%H$_2$O$_2$/10% methanol), permeabilized (0.6% Triton X-100), blocked (1.5% BSA and 1.5% goat serum), and incubated with rabbit anti-ZsGreen (Clontech, Cat. No. 632474, 1:10,000) for 15–18 hr at room temperature. Next, slides were incubated in goat anti-rabbit polyclonal biotinylated conjugated antibody (Vector Labs, Cat. No. BA-1000, 1:500) for 2 hr, followed by incubation with avidin-biotin complex kit (Thermo Fischer Scientific, 32020) for 2 hr and visualization with a 3,3'-diaminobenzidine kit (Vectastain, Vector Labs, SK-4100). Sections were then counterstained with cresyl violet for Nissl labeling.

For co-localization studies, sections were labeled free-floating, 8–10 sections/mouse. Sections were washed with PBS, permeabilized with 0.3% Triton-X, then permeabilized and blocked with 0.3% Triton-X and 5% goat serum (Jackson Immuno). For primary antibodies, the following were incubated for 15–18 hr at 4˚C in PBS with 0.3% Triton-X and 5% goat serum: mouse biotin conjugated anti-NeuN B-clone A60 (Millipore, MAB377, 1:500), rabbit anti-calbindin (Sigma-Aldrich, C2724, 1:500), rabbit anti-parvalbumin (Abcam, ab11427, 1:1000), rabbit anti-FoxP2 (Abcam, ab16046, 1:1000), rabbit anti-DARPP32 (Abcam, ab40801, 1:500). Following incubation in the appropriate antibody, sections were then incubated for 2 hr in the following secondaries: goat anti-rabbit IgG Alexa Fluor 568 (Invitrogen, A11011, 1:500), Cy5 streptavidin (Invitrogen, SA1011, 1:500), and all sections were counterstained with 4',6-diamidino-2-phenylindole (DAPI) (Thermo Fisher Scientific, 1:1000).

For filled MSNs, sections were processed as discussed below. For staining, sections with filled MSNs were permeabilized with 0.6% Triton-X in PBS, then incubated with DARPP32 antibody in 5% goat serum, 0.6% Triton-X in PBS for 48 hr at 4˚C. Sections were washed then incubated with Cy5 streptavidin (1:500) and goat anti-rabbit Alexa Fluor 568 (1:500) for 24 hr at 4˚C.

## Electrophysiology of MSNs

Mouse brain slice electrophysiology recording was performed as described (*Suter et al., 1999*; *Nisenbaum et al., 1994*; *Dorris et al., 2014*; *Willett et al., 2016*; *Haubensak et al., 2010*). Mice were sacrificed with $CO_2$, after which the descending aorta was clamped and the mice were intracardially perfused with ice-cold oxygenated artificial cerebral spinal fluid (ACSF) containing 126 mM NaCl, 1.6 mM KCl, 1.2 mM $NaH_2PO_4$, 1.2 mM $MgCl_2$, 2.4 mM $CaCl_2$, 18 mM $NaHCO_3$, 11 mM glucose. After the olfactory bulbs and the cerebellum were removed, the brain was transferred to a vibratome stage that contained ice-cold ACSF oxygenated with carbogen (95% $O_2$ balanced with $CO_2$); 200 µm coronal sections were then generated with the vibratome (Leica, VT1000S). Brain slices were immediately transferred to oxygenated NMDG-HEPES recovery solution (93 mM NMDG, 2.5 mM KCl, 1.2 mM $NaH_2PO_4$, 30 mM $NaHCO_3$, 20 mM HEPES, 25 mM glucose, 5 mM sodium ascorbate, 2 mM thiourea, 3 mM sodium pyruvate, 10 mM $MgSO_4$, 0.5 mM $CaCl_2$, 300–310 mOsm, titrated with 10 N HCl to adjust pH to 7.3–7.4) and were allowed to recover for 15 min at 32–34°C. After the recovery portion, brain slices were then transferred to room temperature oxygenated ACSF for 1 hr. After the rest period, recordings were performed in a rig equipped with a MultiClamp 700B, a Digidata 1550A1 (Molecular Devices), and a fluorescent microscope (Olympus BX51) that was used to visualize sections and identify TINs. Patch pipettes were pulled with P-97 Sutter micropipette puller to achieve a resistance of 8–16 MΩ, after which they were filled with an intracellular solution (135 mM potassium gluconate, 5 mM EGTA, 0.5 mM $CaCl_2$, 2 mM $MgCl_2$, 10 mM HEPES, 2 mM MgATP, and 0.1 mM GTP, pH 7.3–7.4, 290–300 mOsm). Recording data were sampled at 10 kHz, filtered at 3 kHz, and analyzed (Molecular Devices, pCLAMP, 10.7). For bystander neurons and neurons in uninfected mice, the classification as an MSN was based on the firing pattern in response to the current injections as described previously (*Schmidt and Perkel, 1998*; *Farries and Perkel, 2000*; *Luo et al., 2001*; *Ding and Perkel, 2002*; *Ding et al., 2003*, *Willett et al., 2016*). For filling experiments, once data from a 200 µm section were recorded, the recorded neurons were filled with 0.6% neurobiotin (Vector Labs, SP1120). Filled sections were then stored in 4% PFA for 24 hr followed by cryopreservation in cryoprotective media until stained as described above.

## Microscopy and quantification

Slides for localization data were imaged on a Leica DMI6000 with a motorized stage, using Leica Application Suite X (LAS X) at 10× magnification. Base background subtraction and white balance was maintained throughout individual cohorts. Image stitching was done automatically through LAS X with a 10% overlap and images stored as Leica image file format (lif). Lif images were then processed on a custom MATLAB code for image transformation onto the Allen Institute reference atlas, followed by semi-automated quantification, as described in *Mendez et al., 2018*. Animals were excluded due to having the same or fewer TINs compared to saline-injected animals (~15 TINs) and via ROUT outlier tests. *Figure 1—figure supplement 1* has all mice included and includes 100% of the Allen atlas for the enrichment index. For co-localization studies, 8–10 sections/mouse were imaged. Images were captured on a Zeiss LSM880 inverted confocal microscope (Marley Microscopy Core, UA). Images were captured with a 20× objective, with 1× PMT zoom, all settings were the same across specific staining runs. Autofocus was used to capture a z-stack that would capture from edge-to-edge of the section.

## Statistics

In *Figure 1*, *Figure 1—figure supplement 2*, data were analyzed with a one-sample t-test to compare the enrichment score to a value of 1, which would indicate a random distribution of TINs in a given region of the brain. For *Figures 2–3* and associated figure supplements, the data were analyzed with an unpaired two-tailed t-test to test for differences between II-Cre and III-Cre infected groups. For *Figure 5* and *Figure 7*, the data were analyzed with a Mann-Whitney U-test. *Figure 5—figure supplement 2* was analyzed with an unpaired two-tailed t-test. *Figure 6* was analyzed with a one-way ANOVA with Tukey's post-test. All statistics were done via Prism statistical software (Graph-Pad v9.1.0).

## Acknowledgements

The authors would like to thank the laboratory of Dr Haijiang Cai for the use of their electrophysiology equipment and initial training in electrophysiology. We would also like to thank Dr David Perkel for helpful discussions about the electrophysiology data. Additional thanks to Patricia Jansma and Doug Cromey for imaging training. Finally, thank you to the entire Koshy lab for helpful discussions.

## Additional information

### Funding

| Funder | Grant reference number | Author |
|---|---|---|
| National Institute of Neurological Disorders and Stroke | F99 NS108514 | Oscar A Mendez |
| National Institute of Neurological Disorders and Stroke | R25 NS076437 | Emiliano Flores Machado |
| Arizona Biomedical Research Commission | ADHS14-082991 | Anita A Koshy |
| National Institute of Neurological Disorders and Stroke | R01 NS095994 | Anita A Koshy |

The funders had no role in study design, data collection and interpretation, or the decision to submit the work for publication.

### Author contributions

Oscar A Mendez, Conceptualization, Software, Supervision, Funding acquisition, Investigation, Methodology, Writing - original draft, Writing - review and editing; Emiliano Flores Machado, Software, Formal analysis, Funding acquisition, Validation, Investigation, Methodology; Jing Lu, Formal analysis, Validation, Investigation, Methodology; Anita A Koshy, Conceptualization, Formal analysis, Supervision, Funding acquisition, Validation, Investigation, Methodology, Writing - review and editing

### Author ORCIDs

Anita A Koshy (ID) https://orcid.org/0000-0001-8705-3233

### Ethics

Animal experimentation: All mouse studies and breeding were carried out in accordance with the Public Health Service Policy on Human Care and Use of Laboratory Animals. The protocol was approved by the University of Arizona Institutional Animal Care and Use Committee. (#A-3248-01, protocol #12-391).

### Decision letter and Author response

Decision letter https://doi.org/10.7554/eLife.67681.sa1
Author response https://doi.org/10.7554/eLife.67681.sa2

## Additional files

### Supplementary files

- Supplementary file 1. Area size and percentages of whole brain for quantified regions.
- Supplementary file 2. p-Values for enrichment scores for the 12 quantified regions.
- Transparent reporting form

### Data availability

Data for Figures 1,2,3,5,6,7 is provided.

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
