## [Decision Letter]

**Acceptance summary:**

In this study Mendez and colleagues map the distribution of *Toxoplasma gondii* in mouse brains, using a new imaging approach to detect neurons that have been injected with parasite proteins. Significant regional differences in parasite tropism, as well as selectivity for different neuronal subtypes were observed. Significantly, parasite-affected neurons exhibited severe defects in electrophysiology and may subsequently die, while neighboring uninfected neurons display only minor changes in physiology. These findings highlight the extent to which Toxoplasma interacts with neurons and suggest that even transient or aborted infections can lead to neuronal clearance. This paper will be of interest to those studying parasite infections and host-pathogen interactions.

**Decision letter after peer review:**

Thank you for submitting your article "*Toxoplasma gondii* injected neurons localize to the cortex and striatum and have altered firing" for consideration by *eLife*. Your article has been reviewed by 2 peer reviewers, and the evaluation has been overseen by a Reviewing Editor and Bavesh Kana as the Senior Editor. The reviewers have opted to remain anonymous.

Essential revisions:

1. The reviewers suggested that the available data indicate that the TINs are dying and that the current title and major conclusions suggesting that they just exhibit abnormal electrophysiology is miss-leading. It is strongly recommended that the authors modify their conclusions (and title) regarding the state/fate of TINs unless additional experimental work can be provided to support the conclusion that the TINs have a distinct electrophysiological state.

2. The finding that bystander TINs had a detectable change in electrophysiology (compared to neurons in uninfected mice) was considered to be a significant finding. Could the authors comment or expand on the significance of the electrophysiological changes observed in the by-stander neurons in infected mice compared to neurons in uninfected mice.

3. Consider normalizing TIN numbers to host nuclei rather than tissue area in considering distribution across tissues.

4. Comment or provide information on the extent to which TINs correspond to successful versus aborted infections.

5. Expand discussion to relate these findings to broader implications for Toxo and other neurotropic infections – in particular, the prevailing view that neurons are long-lived reservoirs for pathogens.

*Reviewer #1 (Recommendations for the authors):*

1. Need to change how the data are normalized in Figure 1. But, in my opinion, these data are not necessary in this manuscript and could be removed.

2. The interpretation that inhibitory neurons are not injected is premature. At the very least, the authors need to acknowledge that these neurons may die rapidly following injection.

3. The authors' difficulty with patch-clamping the TINs suggest that the cells are inherently damaged, and this is confirmed by their finding that there are reduced numbers of TINs 8 weeks post infection. Thus, the electrophysiological changes the authors note in those TINs the authors could path-clamp may simply be due to changes in neuronal health rather than a specific impact on their excitability. While they attempted to address this in Figure 6B, it still remains possible that neuronal cell health is the issue. Perhaps the use of channel blockers coupled with more extensive electrophysiological analyses would help to clarify this issue.

4. This finding that TINs are unhealthy and/or die is not trivial nor pedantic and the authors should be encouraged to delve more into this. A minor comment here is how do the authors know that these changes are not simply due to GFP expression.

5. In this reviewer's opinion, the most significant data here are the changes in bystander cell resting membrane potential. But due to neuronal health issues, the authors should define whether this is a specific response to infection or if these neurons also have defects in neuronal health.

*Reviewer #2 (Recommendations for the authors):*

1. Figure 5 shows significance between neurons in uninfected mice and bystander neurons (Figure 5A). In Figure 6 this data is no longer significant – I'm presuming this is an error or if this data is different from Figure 5 then it needs to be explained. The electrophysiology of TINS does seem drastically different however the simplification of the bystander data may be missing important biology. The equivalent of Figure 5B-G is not shown. Because TINs required hyperpolarization to confirm their neuronal status and to measure anything, it is possible that the electrophysiology data alone is not able to detect more subtle differences in neuronal physiology such as a cell that is in a quiescent state? Immunohistochemistry for apoptotic or necrotic markers would be helpful and potentially revealing if combined with cyst markers?

2. If there are no real differences in TINs between Type II and III can conclusions be made about the cause of neuronal death – presumably it is not strain specific proteins?

3. There is constant speculation that Toxoplasma infection causes neurological disease. This work would definitely suggest that it is not a benign infection and has serious consequences. Although bold statements are not needed for good science there could be more links to the broader biology of neurotropic infections in the discussion as nicely alluded to in the introduction.

---

## [Author Response]

Essential revisions:1. The reviewers suggested that the available data indicate that the TINs are dying and that the current title and major conclusions suggesting that they just exhibit abnormal electrophysiology is miss-leading. It is strongly recommended that the authors modify their conclusions (and title) regarding the state/fate of TINs unless additional experimental work can be provided to support the conclusion that the TINs have a distinct electrophysiological state.

We appreciate this suggestion and have changed the title to: Injection with *Toxoplasma gondii* protein affects neuron health and survival. We hope this new title is acceptable.

2. The finding that bystander TINs had a detectable change in electrophysiology (compared to neurons in uninfected mice) was considered to be a significant finding. Could the authors comment or expand on the significance of the electrophysiological changes observed in the by-stander neurons in infected mice compared to neurons in uninfected mice.

We have amended our discussion to highlight that while the bystander physiology is relatively normal compared to TINs physiology, it is still abnormal. In addition, we have also included a new figure (Supp Figure 6) which shows an expected consequence of the depolarized nature of bystanders; they require less input current to trigger the first action potential. In the discussion, we have noted that such changes in other neurons (e.g., cortical neurons) might lead to an increase in seizures— which is seen in patients with symptomatic toxoplasmosis (ie. congenital and recrudescent infections)— or even the various behavioral changes observed in rodents. We have expanded our discussion to include such possibilities but have stressed that these findings are speculations that require further studies to confirm or negate.

3. Consider normalizing TIN numbers to host nuclei rather than tissue area in considering distribution across tissues.

We appreciate the Reviewer’s pointing out that neuron density may play a role in where we find TINs. For example, the limited number of neuron nuclei in/near white matter tracts is almost certainly what accounts for the “under enrichment” of TINs in white matter tracts (in a sense it is a nice built-in control that our system is working). We have noted this explanation in the results.

We also agree that neuron density could explain why TINs somas are enriched in deeper cortical layers and not in more superficial cortical layers. We will incorporate such considerations into future studies/analyses but feel that such re-analysis is beyond the scope of this paper for several reasons. First, prior *T. gondii* studies that have quantified cyst locations have used region size for normalization (Berenreiterova et al., 2011; Boillat et al., 2020). Thus, by using size ourselves, we could be consistent with these papers and draw connections to them. Second, for most of the major brain regions, relative size correlates with the number of TINs found in those regions, suggesting that size may adequately explain why a certain percentage of TINs is found in those regions (even without accounting for nuclei density). From a statistical standpoint, for both type II and type III infection, size does not correlate with the number of TINs for only two regions beyond the white matter tracts: the cortex and cerebellum. Given that the cerebellum is a relatively large brain region and has a high density of neurons in the granular layer— the lack of TINs here suggests that some unknown factor (such as difference in vascular permeability (Daniels et al., 2017)) rather than area size or neuron density accounts for the cerebellum’s relative resistance to infection. Third, unlike brain region size where the Allen Institute Mouse Brain Atlas is accepted as a standard, there is no accepted standard for neuron density, in part because counts vary widely (reviewed in Keller et al., 2018, Tables 1-4). Given the lack of standard counts, the most appropriate way to normalize would be to do our own counts of host nuclei in each region and ideally in both uninfected and infected mice. Counting host nuclei in infected tissue becomes complicated because of the infiltrating immune cells and dividing glia, an issue that cannot be solved by counting only neurons because of a loss of antigenicity for multiple neuron markers in inflamed brain tissue (Cekanaviciute et al., 2014; David et al., 2016). Pursuing such counts in uninfected tissue alone leads to a technical issue for us: when trying to count cells with high numbers using our program (Mendez, Potter et al., 2018), our processing system cannot not handle the workload (i.e., the program crashes). Thus, at this time, we feel that normalizing by size is an appropriate first step and will use the recommended normalization in subsequent work which will pursue defining the mechanisms that lead to enrichment of TINs in the cortex and a lack of TINs in the cerebellum.

4. Comment or provide information on the extent to which TINs correspond to successful versus aborted infections.

With the current Cre system, in vivo, we cannot distinguish between aborted invasion and invasion followed by intracellular killing of the parasite. In addition, as many cysts are in distal neuronal processes (Cabral, Tuladhar et al., 2016), we often cannot determine if a TIN is infected unless we do a full neuron reconstruction, which requires thick sections rather than the 40 micron sections we used for the immunohistochemical studies. For this reason, we do not make distinctions about infection status or how an uninfected but injected TIN arose. We clarified this issue in the text (i.e. TINs refers to both infected and uninfected, injected neurons) as well as included a new figure that more clearly explains this concept. In addition, we have noted that our prior work suggests that >90% of these TINs are not infected, which is consistent with our findings in thick sections where we have done whole neuron reconstructions (unpublished data.)

5. Expand discussion to relate these findings to broader implications for Toxo and other neurotropic infections – in particular, the prevailing view that neurons are long-lived reservoirs for pathogens.

We have expanded the final paragraphs of our discussion to address this concern. We hope we have adequately addressed the comment.

Reviewer #1 (Recommendations for the authors):1. Need to change how the data are normalized in Figure 1. But, in my opinion, these data are not necessary in this manuscript and could be removed.

Please see our response to Essential revisions, point 3.

2. The interpretation that inhibitory neurons are not injected is premature. At the very least, the authors need to acknowledge that these neurons may die rapidly following injection.

We have included this language in the results and discussion of the revised manuscript.

3. The authors' difficulty with patch-clamping the TINs suggest that the cells are inherently damaged, and this is confirmed by their finding that there are reduced numbers of TINs 8 weeks post infection. Thus, the electrophysiological changes the authors note in those TINs the authors could path-clamp may simply be due to changes in neuronal health rather than a specific impact on their excitability. While they attempted to address this in Figure 6B, it still remains possible that neuronal cell health is the issue. Perhaps the use of channel blockers coupled with more extensive electrophysiological analyses would help to clarify this issue.

We have not pursued these studies but agree that future studies with CCB may be useful. As noted above, we have amended our language in the revised manuscript to more clearly define that we cannot say what is driving the ephys changes (neuron health vs. parasite manipulation vs. the immune response, etc).

4. This finding that TINs are unhealthy and/or die is not trivial nor pedantic and the authors should be encouraged to delve more into this. A minor comment here is how do the authors know that these changes are not simply due to GFP expression.

We do not think GFP expression is the issue for several reasons. Many people now identify specific neuron populations by fluorescent protein expression and such issues have not been noted. In addition, as we were concerned that inflammation plus fluorescent protein expression might cause an issue, we pursued patch-clamping of cortical neurons in infected thy1-YFP mice (Jax 003709; Feng et al., 2000). We did not find TINs-like electrophysiology abnormalities in the YFP^+^ cortical neurons (new Figure 6- supplemental figure 2).

5. In this reviewer's opinion, the most significant data here are the changes in bystander cell resting membrane potential. But due to neuronal health issues, the authors should define whether this is a specific response to infection or if these neurons also have defects in neuronal health.

This issue of bystander health is much more difficult to address because it is difficult to track bystanders over time. Several studies by different labs shed light on some of this complexity. Hermes et al., 2008 found no differences in neuronal density at 1 year post infection. Parlog et al., 2014 identified subtle region-specific differences on MRI and showed a mild amount of neuron degeneration (which might be TINs death or bystander death), but most neurons were intact. Of note, both referenced studies used a a type II *T. gondii* strain that is known to cause more inflammation and result in higher cyst numbers compared to the type II strain we used. For these reasons, it is unclear how well those studies would correlate to ours. Finally, while NeuN, MAP2, and β-III tubulin staining distinctly decreases in infected mice (Cekanaviciute et al., 2014, David et al., 2016), nissl staining suggests these neurons are still intact (David et al., 2016). This discrepancy between nissl staining and other neuron-specific antibodies in the setting of inflammation is consistent with the idea that neurons in the inflamed brain can lose some antigenicity— the significance of which we currently do not know. We continue to ponder ways in which to answer this important question.

Reviewer #2 (Recommendations for the authors):1. Figure 5 shows significance between neurons in uninfected mice and bystander neurons (Figure 5A). In Figure 6 this data is no longer significant – I'm presuming this is an error or if this data is different from Figure 5 then it needs to be explained. The electrophysiology of TINS does seem drastically different however the simplification of the bystander data may be missing important biology. The equivalent of Figure 5B-G is not shown. Because TINs required hyperpolarization to confirm their neuronal status and to measure anything, it is possible that the electrophysiology data alone is not able to detect more subtle differences in neuronal physiology such as a cell that is in a quiescent state? Immunohistochemistry for apoptotic or necrotic markers would be helpful and potentially revealing if combined with cyst markers?

Thank you for catching that error; we have addressed it. Of note, given that Figure 5 and Figure 6 use different statistical tests (Figure 5 has 2 groups, Figure 6 has 3), the p-values are different.

In regard to the death of TINs, we are actively pursuing these studies (in a COVID delayed manner) but have nothing to report at this time in terms of cell death in relation to infection and/or the type of cell death the TINs are undergoing. Our preliminary data suggests TINs do not co-localize with staining for cleaved Caspase 3 but these data are too preliminary for publication. Staining for RIPK1 and MLKL are best described in cell culture and are affected by fixative (Samson et al., Cell Death and Differentiation 2021). Unfortunately, the fixative that works best (methanol), kills the GFP signal in such a way that even post-fixation anti-GFP (really antiZsGreen) antibodies (which work with PFA fixed tissue), no longer identify the GFP^+^ cells (we’ve tried). The RIPK3 antibody maybe a possibility but we have not started those experiments yet. Trying to antibody stain thick tissue (which is what would be needed to define if the TIN is infected or not) will then represent the next hurdle. In short, we are using multiple methods to try to address this question, but, currently, we have no reliable data to address the Reviewer’s comment.

2. If there are no real differences in TINs between Type II and III can conclusions be made about the cause of neuronal death – presumably it is not strain specific proteins?

As the Reviewer noted, as we see TINs decrease in both type II and type III infection (type II is anecdotal- we have not performed carefully matched studies with the type II parasites at this time), we do not think these outcomes are strain specific. We have not included this statement in our discussion because, like most journals, *eLife* prefers we not reference such anecdotal data.

3. There is constant speculation that Toxoplasma infection causes neurological disease. This work would definitely suggest that it is not a benign infection and has serious consequences. Although bold statements are not needed for good science there could be more links to the broader biology of neurotropic infections in the discussion as nicely alluded to in the introduction.

We appreciate the Reviewer’s thoughtful comments. While we agree that the loss of neurons would not be considered benign, an actual global/functional consequence of losing these neurons remains to be shown. As the consequences of Toxoplasma infection— especially in humans— has been overstated by the media compared to the data (please see Johnson, Koshy mBio 2020), we have refrained from making statements about the nature of this neuronal loss. We have expanded our discussion of how our data might relate to behavior changes in rodents (bystanders) and affect how we think about neurotropic infections. We hope the Reviewer will feel that we sufficiently “connected the dots” for how our findings have important and broad implications. Please also see our response to Essential revisions, points 2 and 5.